# Femtosecond Laser Drilling of Cylindrical Holes for Carbon Fiber-Reinforced Polymer (CFRP) Composites

**DOI:** 10.3390/molecules26102953

**Published:** 2021-05-16

**Authors:** Hao Jiang, Caiwen Ma, Ming Li, Zhiliang Cao

**Affiliations:** 1State Key Laboratory of Transient Optics and Photonics, Xi’an Institute of Optics and Precision Mechanics, Chinese Academy of Sciences, Xi’an 10068, China; jianghao@opt.ac.cn (H.J.); caozhiliang@opt.ac.cn (Z.C.); 2University of Chinese Academy of Sciences, Beijing 100049, China

**Keywords:** femtosecond laser drilling, CFRP, HAZ, cylindrical hole, spiral drilling apparatus

## Abstract

Ultrafast laser drilling has been proven to effectively reduce the heat-affected zone (HAZ) of carbon fiber-reinforced polymer (CFRP) composites. However, previous research mainly focused on the effects of picosecond laser parameters on CFRP drilling. Compared with a picosecond laser, a femtosecond laser can achieve higher quality CFRP drilling due to its smaller pulse width, but there are few studies on the effects of femtosecond laser parameters on CFRP drilling. Moreover, the cross-sectional taper of CFRP produced by laser drilling is very large. This paper introduces the use of the femtosecond laser to drill cylindrical holes in CFRP. The effect of laser power, rotational speed of the laser, and number of spiral passes on HAZ and ablation depth in circular laser drilling and spiral laser drilling mode was studied, respectively. It also analyzed the forming process of the drilling depth in the spiral drilling mode and studied the influence of laser energy and drilling feed depth on the holes’ diameters and the taper. The experimental results show that the cylindrical hole of CFRP with a depth-to-diameter ratio of about 3:1 (taper < 0.32°, HAZ < 10 μm) was obtained by using femtosecond laser and a spiral drilling apparatus.

## 1. Introduction

Carbon fiber reinforced polymer (CFRP) composites have the characteristics of high strength-to-weight ratio, high modulus-to-weight ratio, lower density, and good thermal stability. They have been widely used in aerospace and other fields [1]. CFRP is generally formed by mixing and solidifying fiber filaments and epoxy resin, laying and laminating according to different laying directions [2]. Because their performance is mainly affected by the characteristics of carbon fiber, epoxy resin, and the orientation of the fiber, they have low interlayer strength, anisotropy, high hardness, and high brittleness, which makes CFRP traditional machining prone to a series of problems, such as carbon fiber pull-out, delamination, excessive tool wear, abrasive penetration, and abrasive slurry disposal [3,4,5].

Laser drilling is a non-contact stress processing method; it can be used for riveting and bolting to assemble composite laminates with other components, and it is considered to be a CFRP processing method with very broad prospects for application [6]. However, due to the difference in the thermal expansion coefficients of carbon fiber in the radial and longitudinal directions and the difference in the thermal properties of carbon fiber and resin, it is difficult to process CFRP by laser. The main manifestation is that there are thermal damage defects, including a heat-affected zone (HAZ), fiber pull-out, and material separation after laser processing [7]. The thermal effect can be reduced by increasing the processing speed [8], reducing laser-material interaction time [9], using high-quality lasers [10], or reducing the laser power [11]. Among them, reducing the laser-material interaction time, especially the use of ultrafast lasers to process CFRP, has been proven to be an important means to effectively suppress thermal effects.

Freitag et al. [12] used a 1.1 kW infrared picosecond laser to cut 2 mm thick CFRP and obtained a thermal damage size of less than 20 μm; the effective cutting speed reached 0.9 m/min, which fully demonstrated the advantages of ultrafast laser processing CFRP. Salama et al. [13] used a 400 W infrared picosecond laser to drill 6 mm thick CFRP and realized a HAZ < 25 μm. They found that as the laser power decreases and the scanning speed increases, while the HAZ size and etching depth decrease. Hu et al. [14] used the Nd: YVO4 picosecond pulse system to conduct experimental research on CFRP cutting, and achieved a HAZ size of 13~44 μm at a scanning speed of 1500 mm/s. The study showed that HAZ has a minimum value at lower laser power and larger hatch distance. Compared with the picosecond laser, the femtosecond laser has a smaller pulse interval, which can realize the processing of carbon fiber materials with higher precision and less thermal effect [15]. Keiji SONOYA et al. [16] used sapphire femtosecond lasers to cut FRP, CFRP, and PC samples. The results showed that the fiber damage of FRP and CFRP was slight and that a smooth cut surface was obtained. Yongdu Li et al. [17] used a UV femtosecond laser to cut carbon fibers with a thickness of 2 mm, obtaining a HAZ of about 25 μm. They also used a single factor method to analyze the influence of laser average power, repetition frequency, and scanning speed on the cutting quality. However, there are few research studies on the influence of femtosecond laser parameters on drilling quality and HAZ of CFRP.

In addition, laser drilling of CFRP is mainly realized at present by scanning galvanometer [13,18,19]. Due to the limitation of the angle between the beam and the material, this drilling method can only achieve holes with large taper in cross section. In order to realize the cylindrical hole drilling of CFRP, it is necessary to use a special drilling device, such as spiral drilling apparatus [20] or helical drilling head [21]. However, the drilling principles of these drilling devices and the scanning galvanometer are completely different, resulting in a large difference in the mechanism of laser drilling on CFRP; thus, investigating in great detail the laser drilling cylindrical holes in CFRP is a matter of great urgency.

This paper reports the use of femtosecond lasers and spiral drilling apparatus to drill cylindrical hole in CFRP. The main purpose is to study the effect of laser power, rotational speed of laser, and number of spiral passes on HAZ and ablation depth in circular laser drilling and spiral drilling mode, as well as to study the influence of laser energy and drilling feed depth on the holes’ diameters and the taper in spiral drilling mode, so as to finally realize the drilling of cylindrical holes in CFRP with small HAZ.

## 2. Materials and Experimental Setup

CFRP with a thickness of 2 mm was used in this study and consisted of directional 0° and 90° lay-ups of carbon fiber. The experimental setup used for this study included a femtosecond laser, a beam expander, a λ/4 plate, a reflector, a focus lens, a three-axis motion platform, and a controller, as shown in Figure 1. The femtosecond laser was produced by PHAROS. It delivers a 290 fs width pulse with a maximum pulse energy of 200 µJ and works at a wavelength of 1028 ± 5 nm with a maximum repetition rate of 1100 kHz. The laser has an adjustable power range from 1 W to 10 W, and the power density distribution of the laser beam is a Gaussian distribution with a beam quality factor (M^2^) of approximately 1.2. The main parameters of the laser used in the experiments are listed in Table 1.

After the beam emitted from the laser, the beam expander expanded the beam in order to gain a fine focusing spot; after that, the beam went through the wave plate to ensure circular polarization of the beam before the reflector. Thereafter, the beam passed through a new spiral drilling apparatus (Xi’an Institute of Optics and Precision Mechanics, Chinese Academy of Sciences, SpiralDrilling-1) and focus lens (Xi’an Institute of Optics and Precision Mechanics, Chinese Academy of Sciences, focal length:100 mm), focusing on the sample surface (focal spot: 21 μm).

The spiral drilling apparatus consisted of a plate glass (thickness of 16 mm) and double optical wedges (wedge angle of 0.35°); therefore, the spatial attitude adjustment of the laser could be realized through the combination of these optical elements. Furthermore, by the rotation of these optical elements and the dynamic adjustment of the relative position, the laser could realize circular laser drilling trajectories or spiral laser drilling trajectories and apply them on the sample surface at an adjustable angle. The specifications of the spiral drilling apparatus are listed in Table 2.

A three-axis motion platform in the experimental system was used to realize the movement of the sample in the XY direction and the feeding of the spiral drilling apparatus in the Z direction; the controller was used to send the commands of the motion platform. In this study, the sample was fixed on the XY moving platform with a clamp, and the processing position was adjusted through the XY moving platform. Then, the Z-moving platform drove the spiral drilling apparatus to carry out layer by layer feeding with a fixed distance, so as to realize the cylindrical holes drilling. In the experiment, we firstly compared the changes in the drilling depth and HAZ of the circular laser drilling and spiral laser drilling trajectories at different rotation speeds and powers (as shown in Figure 2). Then we studied the influence of different passes of the spiral laser drilling trajectory on the drilling depth and HAZ. Finally, we studied the taper change of the holes at different feeding depths so as to obtain the femtosecond laser drilling parameters of the cylindrical hole.

All experiments in this study were carried out in air, and compressed air with a pressure of 0.4 MPa was used in the whole drilling process to remove the debris and plasma generated during the processing. After that, the samples were cleaned with compressed air, and the size of the HAZ and depth of holes were studied using a confocal microscope (VK-X1000, Keyence, Osaka, Japan); the size of holes was measured by a digital microscope (VHX-950F, Keyence, Osaka, Japan).

## 3. Results and Discussion

### 3.1. Effect of Laser Power and Rotational Speed of Laser on HAZ and Ablation Depth

#### 3.1.1. Circular Laser Drilling Trajectories

For circular laser drilling trajectories, HAZ and removal depth were investigated using different laser power and rotational speed. Since the laser ablation threshold of CFRP is 0.3 J/cm^2^ [13], the laser powers in this experiment were set as 0.57 W, 0.7 W, 1.39 W, and 2.08 W under 50 K repetition rate, for which the corresponding laser fluences were 3.3 J/cm^2^, 4 J/cm^2^, 8 J/cm^2^, and 12 J/cm^2^ according to Equations (1) and (2), where *R* is repetition rate of laser, *W* is laser power, *P* is the laser fluence, *E* is the single pulse energy, and *D* is the diameter of the laser spot.

(1)$$E=\frac{W}{R}$$
(2)$$P=\frac{4E}{\pi D^{2}}$$

Figure 3a,b show that the ablation depth and HAZ of CFRP were reduced as the rotational speed of the laser increased, while both increased for higher power. As the rotational speed of the laser increased, the number of laser pulses per unit area upon on the material decreased, which led to the ablation depth and HAZ decreases, a result that is consistent with previous studies [18,22]. It is worth noting that in Figure 3a, the ablation depth corresponding to the laser power of 0.7 W to 2.08 W does not change significantly with the increase of rotational speed of laser between 600 rpm to 1800 rpm. This is because when using single loop drilling, the kerf width is small, resulting in debris or plasma that can prevent the laser beam from reaching the material further—debris or plasma that cannot be removed well by compressed gas.

#### 3.1.2. Spiral Laser Drilling Trajectories

Figure 4a,b show the influence of different power/rotational speeds on drilling depth and HAZ with spiral laser drilling trajectories. It took 25 passes as a cycle, which represents the number of passes from the largest ring to the smallest ring. It can be seen from Figure 4 that change in laser power and rotational speed of laser have a similar influence on drilling depth and HAZ as those of circular laser drilling trajectories. The main difference is that under the same laser power, the depth of spiral laser drilling trajectories is about 10 times larger than that of circular laser drilling trajectories, and the HAZ is smaller or not (see inset of Figure 4b, the HAZ of the hole top surface). This indicates that spiral laser drilling of CFRP with femtosecond lasers has obvious advantages in reducing the HAZ size compared with laser drilling using picosecond or other longer pulse lasers [12,23]. By spiral laser drilling, wider cavities are formed inside the material. These cavities can effectively prevent non-ejected material such as plume from blocking the laser drilling, resulting in more material removal. At the same time, spiral drilling can achieve a more uniform distribution of laser power on the surface of the hole, reducing the interaction time of the laser pulse at the same position, thus reducing the HAZ [24].

In addition to the above, it is worth noting that the depth variation of auger drilling was not uniform. The spiral drilling apparatus in this study rotated at a certain speed leading the rotating beam to form a spiral track at the same angular speed, which is different from a previous study in which a galvanometer was used to drill at an equal linear speed [13,18,19]. Therefore, the cross-section profile of the hole formed by the device in the drilling process was also different from that of drilling with a galvanometer. In addition, the spacing of spiral was also not even according to Equation (3), which describes the relationship between the relative angle of the double optical wedges in the spiral drilling apparatus and the beam rotation radius (in Equation (3), *r* is the beam ring radius, *f* is the focal length, *α* is the optical wedges angle, *n* is the optical wedges refractive index, *φ* is relative rotation angle between the double optical wedges).
(3)r=2f(n−1)αcosϕ

In order to obtain the change rate of the beam ring radius of spiral drilling, both sides of Equation (3) are derived from *φ*, as shown in Equation (4).
(4)$$\frac{dr}{d\phi }=-2f(n-1)\alpha \sin\phi$$

Figure 5 shows that the change rate of r relative to *φ* (express as *dr*/*dφ* in Equation (4)) under different angles between the double optical wedges (α=0.0061, *n* = 1.458462, *f* = 100 mm, $0\leq \phi \leq \frac{\pi }{2}$; additionally, ϕ=0 and $\phi =\frac{\pi }{2}$ marked with green circles, respectively, and the relative angle of double optical wedges illustrated in Figure 5, where ϕ=0 represents the maximum radius of the spiral laser drilling trajectories, and $\phi =\frac{\pi }{2}$ represents the minimum radius of the spiral laser drilling trajectories). It can be seen that with the increase of the angle between the double optical wedges, *dr*/*dφ* also increases, which indicates that the distance between the rings changes from smaller to wider when the laser is drilling from the outer ring to the inner ring.

According to the above principles, the depth distribution of cross-section profile in the process of spiral laser drilling was studied. Figure 6 shows the depth distribution along diameter of the hole measured by confocal microscope. The change of depth can be divided into a non-central region (indicated by yellow lines in Figure 6) and a central region (indicated by red lines in Figure 6). In the non-central region, the depth changes slowly, but in the central region, the depth changes rapidly. One possible explanation is that in the central region of the hole, although the spiral spacing is wider, the diameter of the ring is very small, resulting in the overlapping area between the spots that irradiates on the ring being much larger than that in the non-central region, so more laser power irradiates on the central region of the hole, resulting in more material removal in this region. (see Figure 7)

In the non-central area along the diameter direction, with the increasing of the diameter of the spiral laser drilling trajectories, the distance between the spiral rings decreases, which leads to the increase of the spot overlap rate between the rings, which means more material will be removed. At the same time, because the spiral drilling apparatus rotates at the same angular speed during processing, the spot overlap rate of the ring with larger diameter is smaller than that of the ring with smaller diameter, which means there will be less material removed (see Equation (5), where *R* is the laser repetition rate, *d* is ring diameter, *v* is angular speed of laser drilling, and *q* is spacing of laser spots).
(5)$$q=\frac{R}{dv}$$

As a result, under the combined effect of these two factors, the depth of the non-central area changes slowly. This change process means that the central region of the hole will be the first to penetrate, providing a channel for the escape of debris and plasma (especially for small holes with tens of microns or even hundreds of microns) and avoiding the formation of positive taper holes due to debris and plasma blocking the laser. Consequently, it is very beneficial for cylindrical hole drilling.

### 3.2. Effect of Number of Passes on HAZ and Ablation Depth Using Spiral Drilling

Based on the characteristics of spiral drilling, the effect of the number of passes on HAZ and ablation depth was studied. In this experiment, 25, 30, 35, and 40 passes were selected as a spiral drilling cycle. Figure 8 shows that in a cycle, the greater the number of spiral passes, the greater the hole depth (50 KHz, 0.7 W, 2400 rpm). This is due to the fact that in one cycle of spiral drilling, the more passes, the more the laser power per unit area acting on the inside of the material, leading to more material removal.

However, the HAZ produced by different number of passes is very small, which can hardly be observed under confocal microscope, as shown in Figure 9. This shows that under laser power of 0.7 W and 2400 rpm rotational speed of the laser, the change of the number of passes in one cycle is not enough to have a greater impact on HAZ. It can also be attributed to the fact that when using spiral laser drilling, the cumulative laser irradiation time on CFRP is greatly shortened, thus inhibiting the generation of HAZ. In addition, it can be seen that the change of spiral spacing has much less influence on the HAZ than the change of laser power. Furthermore, it also shows that a femtosecond laser has more obvious advantages than a picosecond or longer pulse laser in HAZ suppression [15]. However, it can be predicted that after a long time of laser irradiation, the HAZ generated by the drilling with fewer cycles will be better than that with more cycles according to Figure 3b. Moreover, if more passes are used in the spiral laser drilling cycle, the drilling efficiency can be improved due to more power deposited in a unit area of the processing zone, but more passes are not good for reducing HAZ. On the contrary, fewer passes are beneficial to HAZ, but it will reduce the drilling efficiency.

### 3.3. Effect of Laser Power and Drilling Feed Depth on Hole Diameter and Hole Taper

In this study, layer by the layer feeding method was adopted to realize the through-hole processing; the feed amount of each layer was 0.01 mm with 5 s dwell time, and the whole drilling process was assisted by 0.4 MPa coaxial compressed air. Figure 10 shows the variation of the entrance and exit holes’ diameters obtained by using different laser power and drilling feed depth on 2 mm thick CFRP. When the laser power is 0.7 W, the entrance diameters are larger than the exit diameters, which means that the holes cross section are positive cone, and this trend basically remains unchanged at different drilling feed depths, as shown in Figure 10a. When the laser power is increased to 1.39 W, the diameters of the exit holes increase significantly (Figure 10b). Especially when the laser power is 2.09 W, the diameters of the exit holes even exceed the entrance holes, as shown in Figure 10c. In addition, it can be found that with the increase of laser power, the entrance and exit diameters also increase, in which the entrance diameter gradually increases with a slow trend, and the rate of increase of the exit diameter is much faster than that of the entrance diameter. According to the electron lattice double temperature model [25], the laser power is proportional to the temperature of the electron and the lattice. When the laser power increases, the material can absorb more laser power, resulting in a violent lattice phase explosion process, which makes the pressure generated by the material vapor and high-power particle jet stronger, and eventually leads to more intense ablation of the material, leading to an increase of the diameter. Furthermore, because the laser power obeys Gaussian distribution, when the power density increases, only the effective laser spot will increase at the entrance of the hole. At the same time, more power is applied to the inside of the hole, allowing more power to be absorbed by the material inside the hole. As a result, the effect of higher laser power on the exit of the hole is greater than that at the entrance of the hole [26]. In addition, with the increase of drilling depth, the exit diameters also increase. This can still be attributed to the power absorption of the internal material. With the increase of drilling feed depth, the ablation power acting on the bottom material increases correspondingly, and the material is removed more thoroughly, which leads to the increase of the holes’ exit diameters.

Furthermore, Figure 10c shows that the bottom aperture has exceeded the front aperture when the laser power is 2.08 W. This phenomenon is mainly attributed to two factors. The first is that the higher power leads to the increase of the bottom aperture, as described above. On the other hand, there is a certain angle between the laser beam and the surface of the part at this time, which leads to the inverted cone shape of the processed hole. Figure 11 shows that when the double optical wedges in the spiral laser drilling apparatus are rotated at different angles relative to each other, the beams are also in different postures relative to the sample. When the relative rotation angle is 90° (the initial state of the double wedges is shown in Figure 5), the exit angle (expressed by θ) of the beam after the double wedges is 0°, and then the beam is focused and forms a certain angle with the sample surface. At this time, the aperture that can be drilling is the smallest, and the cross-section is an inverted cone, as shown in Figure 11a. When the relative rotation angle at 90°>θ>0°, the diameter gradually increases as the relative rotation angle of the double wedges decreases, and the inverted cone angle of the hole section also gradually decreases, as shown in Figure 11b. Figure 11c shows that when θ = 0°, the diameter is the largest, and the focused beam is perpendicular to the sample surface. In this experiment, the relative rotation angle of the double wedges was 36°, so the beam was in the inverted cone processing state relative to the sample, resulting in the exit diameters shown in Figure 10c being larger than the entrance diameters.

### 3.4. Cylindrical Holes Drilling Realized by Spiral Drilling

By comprehensively considering factors such as HAZ and drilling efficiency, this experiment selected a set of femtosecond laser and spiral laser drilling parameters to realize the processing of cylindrical holes with small HAZ. These parameters were as follows: 1.39 W and 50 KHz as femtosecond laser parameters for laser power and repetition rate, respectively; rotational speed of laser of the spiral drilling apparatus was 2400 rpm, the relative rotation angle of the double wedges was 36°, and the spiral laser drilling cycle was 25 passes. The hole drilling was processed layer by layer; the drilling feed depth was 1.8 mm with 0.01 mm feed and 10 mm/s feed speed for each layer. The dwell time of each layer was 5 s. Figure 12 shows the cross-sectional profile of a cylindrical hole with φ683 μm on 2 mm thick CFRP plate measured by confocal microscope, and the taper of the cylindrical hole is <0.32° under about 3:1 depth-to-diameter ratio.

In addition, the HAZ was measured using a digital microscope at the entrance and the cross section of the hole. Only small HAZs (<10 μm) were found at the entrance, which was better than picosecond laser drilling (<25 μm) [13] or nanosecond laser drilling (<50 μm) [24], and no obvious HAZ at the cross section, as shown in Figure 13 and Figure 14. The HAZ at the entrance of the hole may be caused by the fact that when the laser is fed to the bottom of the hole through the movement of the Z axis, the beam away from the laser focus irradiates the material at the entrance of the hole, and the beam energy of the beam was weak, which cannot remove the material completely. Therefore, most of the energy absorbed by the material was converted into heat, resulting in HAZ. However, the inside of the hole was always near the focus of the laser; by selecting an appropriate femtosecond laser energy, it can be realized that the energy absorbed by CFRP during laser drilling is mainly used for material removal rather than conversion into heat, thereby avoiding heat accumulation.

The femtosecond spiral laser drilling described in this article has obvious advantages over other pulsed lasers and drilling technologies in the high-precision hole drilling of CFRP. Table 3 lists the comparison of lasers with different pulse widths on the HAZ and drilling ability of hole cross-section topography with different drilling devices.

## 4. Conclusions

The work discusses the use of femtosecond lasers and spiral drilling apparatus to realize cylindrical holes with small HAZ drilling on CFRP. From this work, the following conclusions can be drawn:(1)Spiral drilling can greatly reduce the HAZ sizes and increase the drilling depth compared with drilling with circular laser drilling trajectories under the same processing parameters.(2)When using a spiral drilling apparatus, the distribution of the spiral pitch is sparse at first and then dense due to the double optical wedges rotating at a fixed angular velocity. This change causes the hole to penetrate quickly and provides an escape channel for the residue and plasma generated by the drilling.(3)The diameters at the entrance and exit increase with the increase of laser power and drilling feed depth, especially for the diameters at the exit, while the taper of the cross section of the hole changes in an opposite trend.(4)The femtosecond laser and spiral drilling apparatus were successfully used to realize the drilling of a cylindrical hole with a taper < 0.32° under about 3:1 depth-to-diameter ratio. The HAZ size at the entrance hole was <10 μm, and no obvious HAZ was found in the cross section. Therefore, the femtosecond laser spiral drilling method can achieve high-precision drilling, thereby ensuring high joint strength and avoiding any weakening of the CFRP structure.

## Figures and Tables

**Figure 1 molecules-26-02953-f001:**
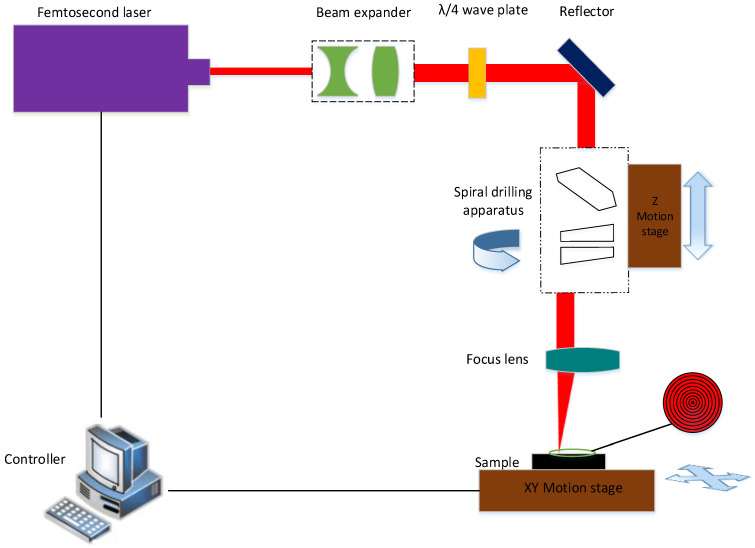
Schematic diagram of the femtosecond laser spiral drilling apparatus.

**Figure 2 molecules-26-02953-f002:**
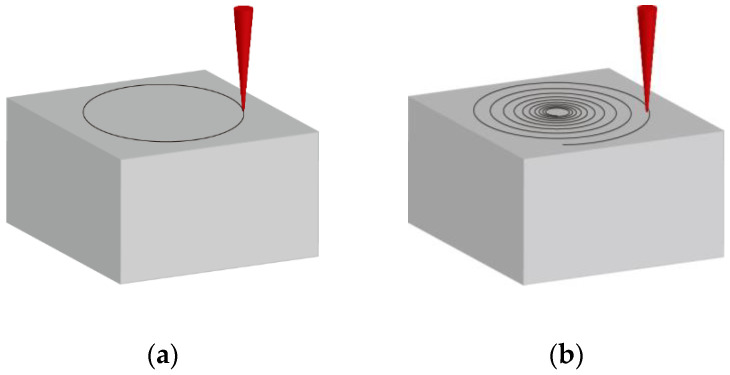
(**a**) Circular laser drilling trajectories and (**b**) spiral laser drilling trajectories.

**Figure 3 molecules-26-02953-f003:**
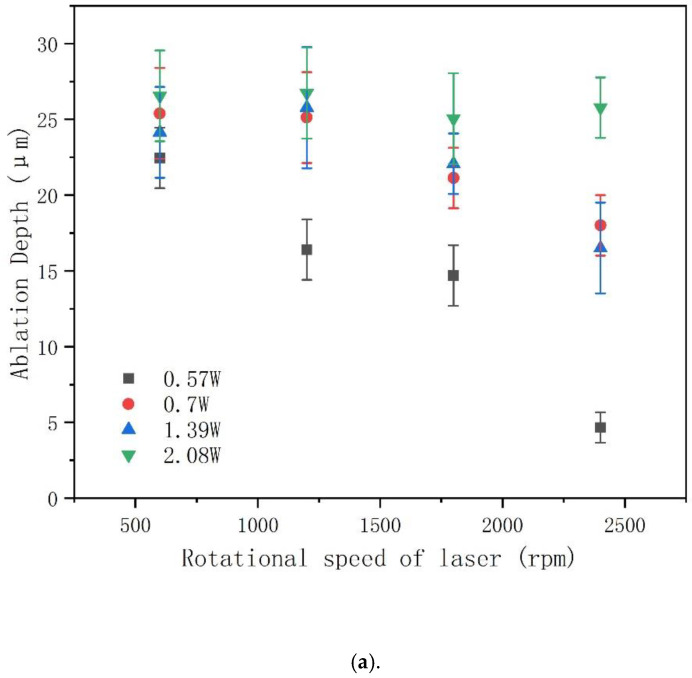
Effect of power/rotational speed on (**a**) ablation depth and (**b**) HAZ using circular laser drilling trajectories. Repetition rate: 50 KHz; powers: 0.57 W, 0.7 W, 1.39 W, and 2.08 W; number of passes: 25 passes; diameter: φ678 um; rotational speed of laser: 600 rpm, 1200 rpm, 1800 rpm, 2400 rpm.

**Figure 4 molecules-26-02953-f004:**
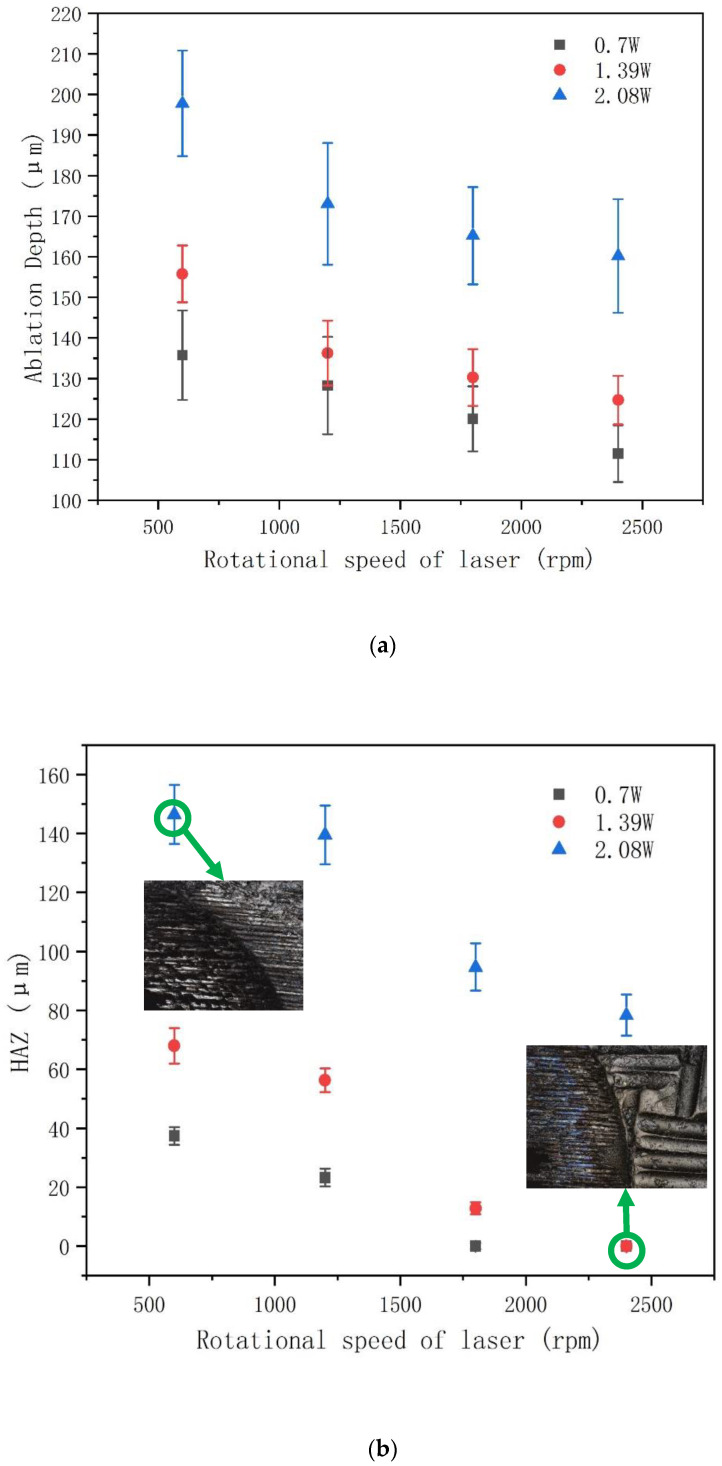
Effect of power/rotational speed on (**a**) ablation depth and (**b**) HAZ using spiral laser drilling. Repetition rate: 50 KHz; powers: 0.7 W, 1.39 W, and 2.08 W; number of passes: 25 passes; diameter: φ 678 um; rotational speed of laser: 600 rpm, 1200 rpm, 1800 rpm, 2400 rpm.

**Figure 5 molecules-26-02953-f005:**
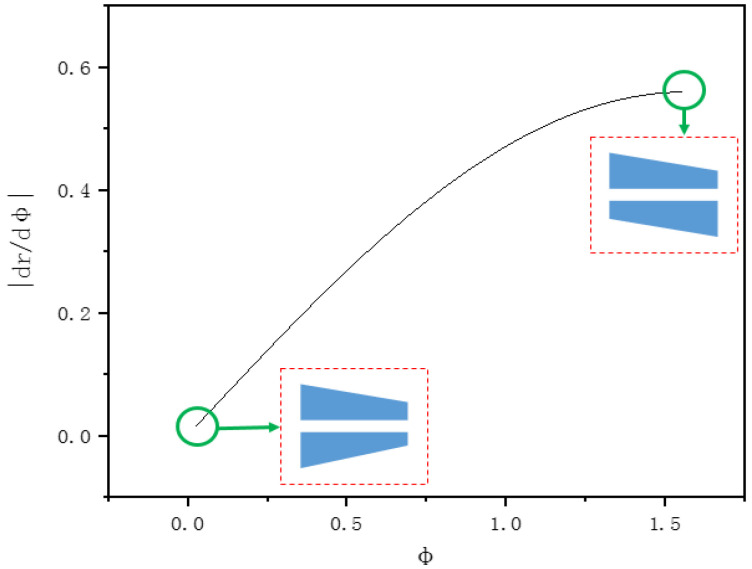
The change rate of r relative to φ under different angles between the two optical wedges.

**Figure 6 molecules-26-02953-f006:**
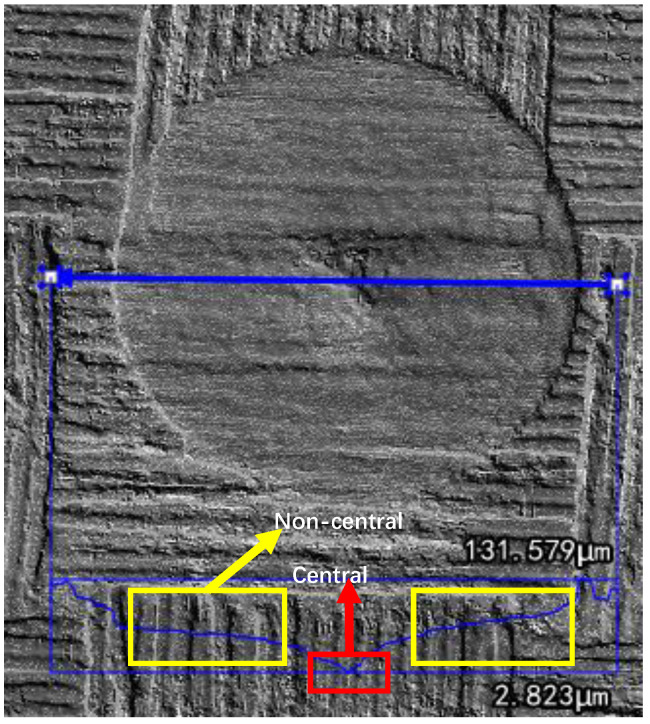
The depth distribution along diameter of the hole measured by confocal microscope.

**Figure 7 molecules-26-02953-f007:**
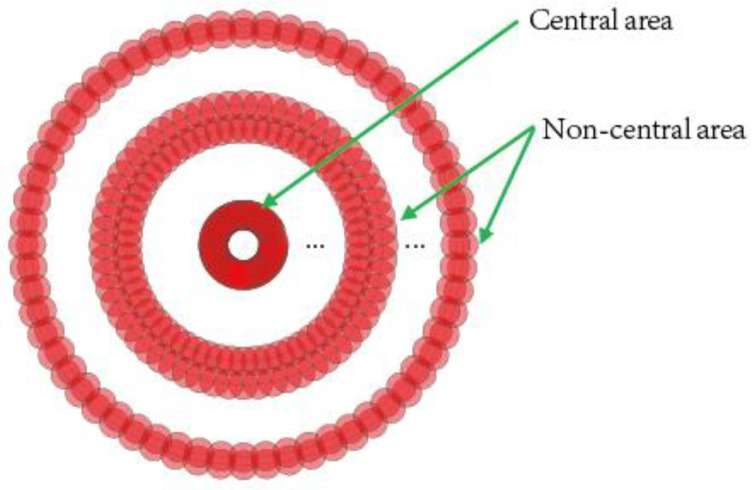
Depth profile of spiral drilling. Laser power: 0.7 W; number of passes: 25; repetition rate: 50 KHz.

**Figure 8 molecules-26-02953-f008:**
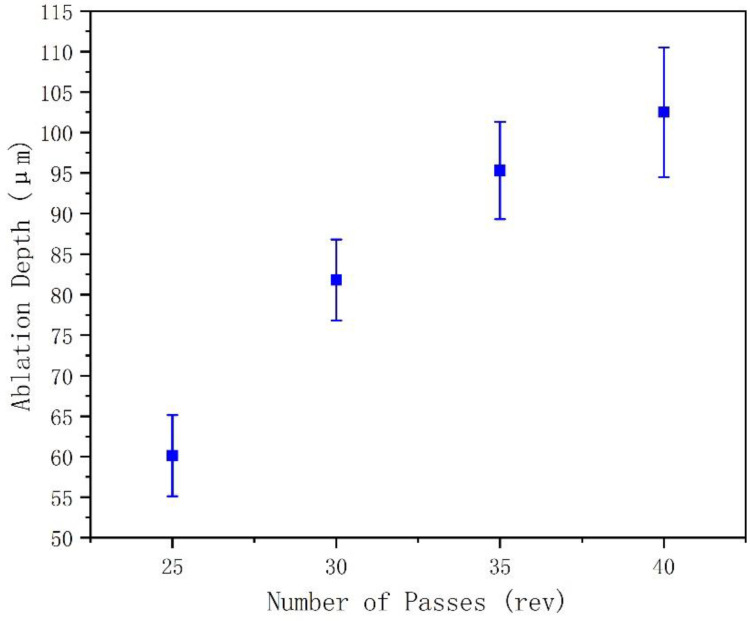
Effect of number of passes on hole-depth in a spiral drilling cycle.

**Figure 9 molecules-26-02953-f009:**
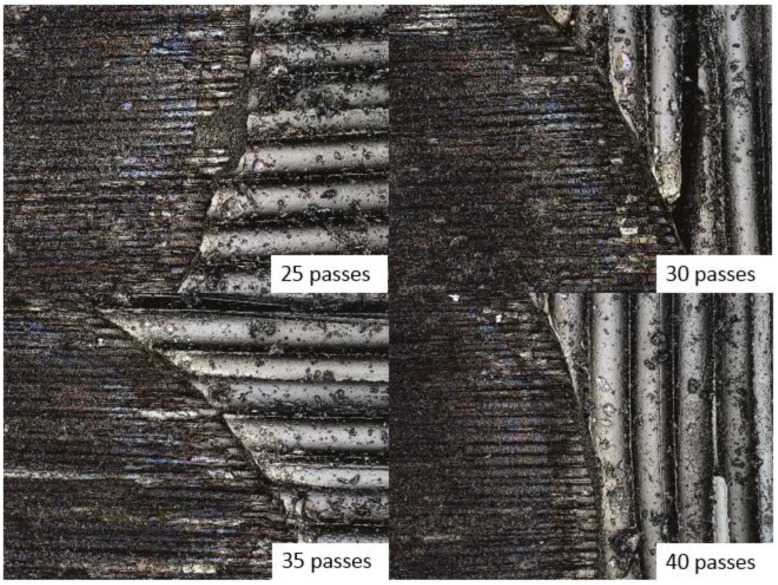
Effect of number of passes on HAZ in a spiral drilling cycle.

**Figure 10 molecules-26-02953-f010:**
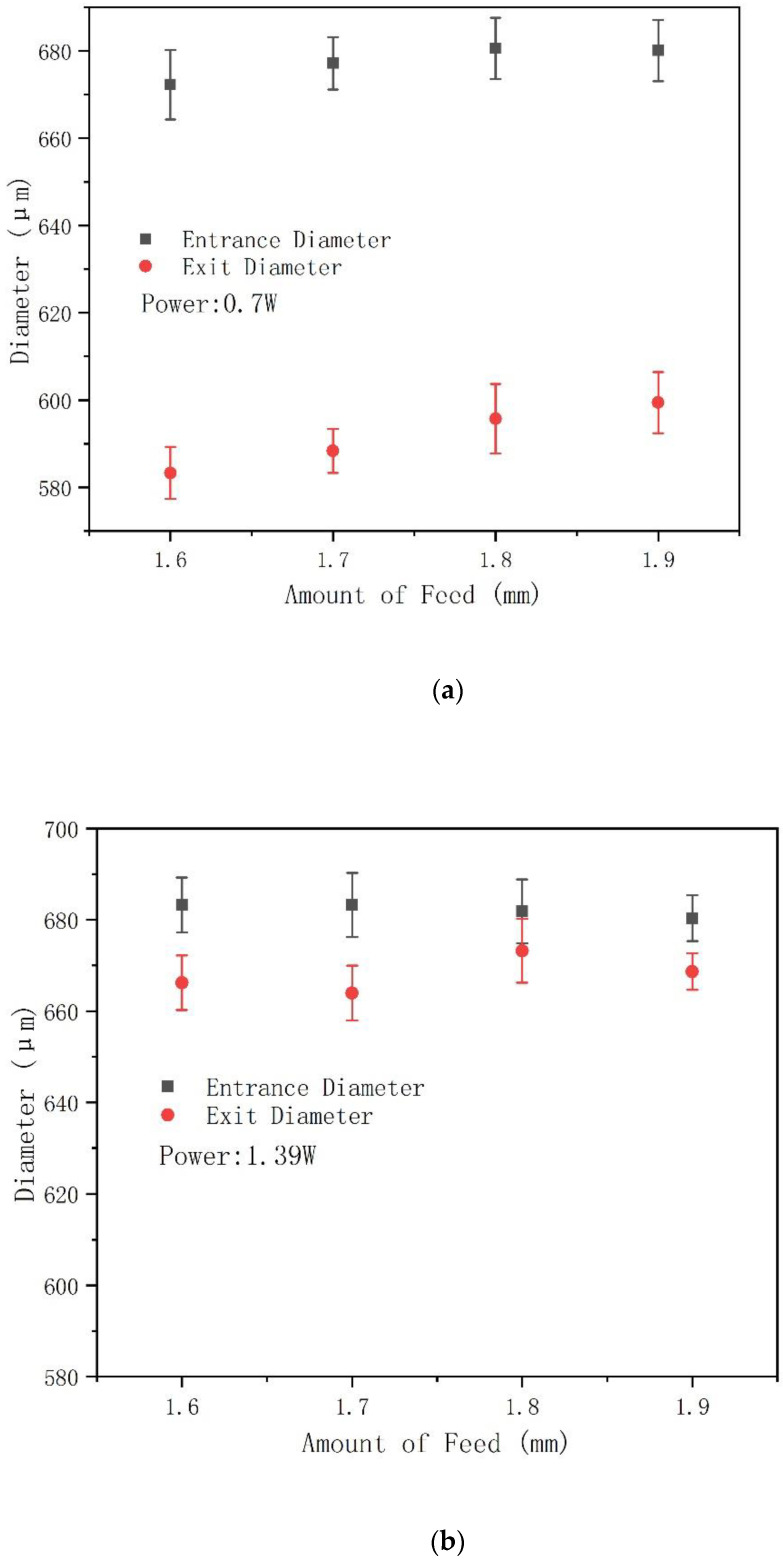
Variation of the entrance and exit holes’ diameters obtained by using different laser power ((**a**) 0.7 W, (**b**) 1.39 W, (**c**) 2.08 W; repetition rate: 50 KHz; rotational speed of laser: 2400 rpm; feed speed of Z movement: 10 mm/s; drilling cycle: 25 passes) and drilling depth (1.6 mm to 1.9 mm) on 2 mm thick CFRP plates.

**Figure 11 molecules-26-02953-f011:**
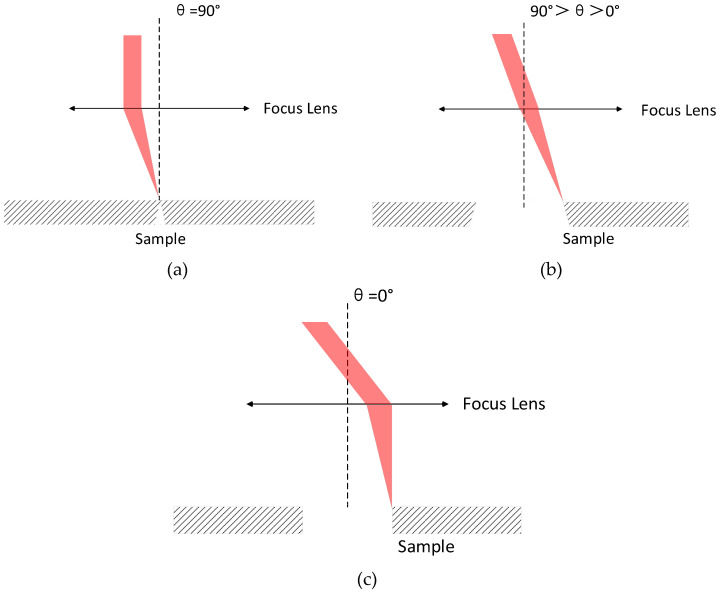
The relationship between the relative rotation angle of the double wedges of spiral drilling apparatus and the beam angle.

**Figure 12 molecules-26-02953-f012:**
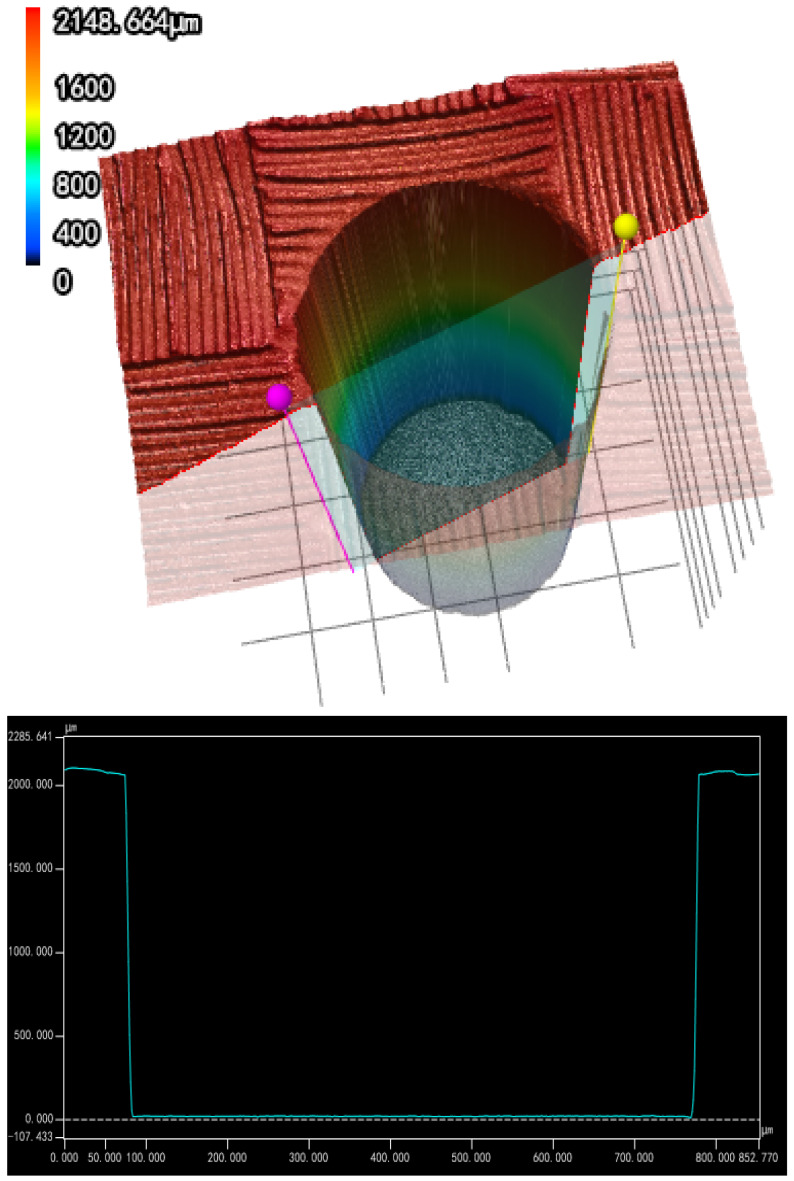
Cross-sectional profile of a cylindrical hole measured by the confocal microscope.

**Figure 13 molecules-26-02953-f013:**
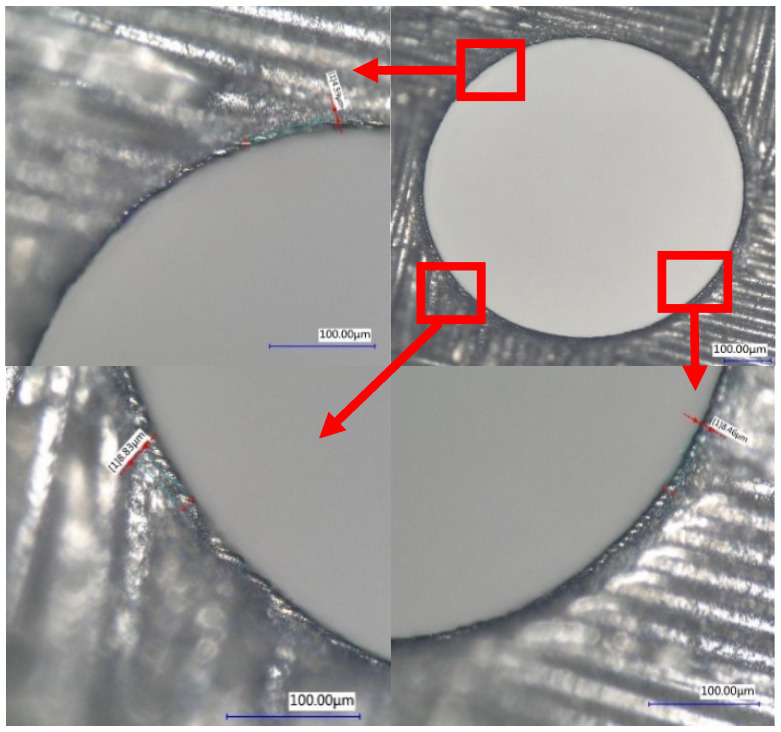
The HAZ size at the hole’s entrance.

**Figure 14 molecules-26-02953-f014:**
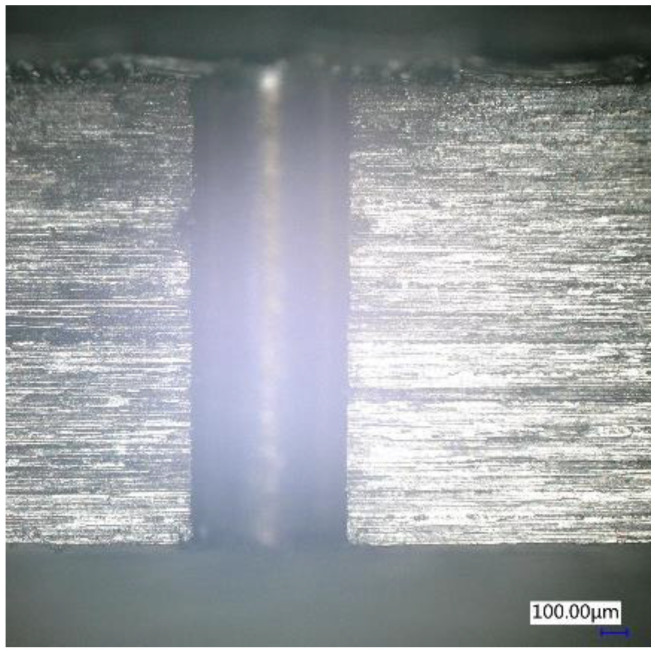
Cross-sectional view of 2 mm drilled hole.

**Table 1 molecules-26-02953-t001:** Parameters of the femtosecond laser.

Laser Parameters	PHAROS-10 W
Max average power	10 W
Pulse duration	<290 fs
Pulse energy	>0.2 mJ
Beam quality	M^2^ < 1.2
Center wavelength	1028 nm ± 5 nm
Beam diameter	1.8 mm

**Table 2 molecules-26-02953-t002:** Specifications of the spiral drilling apparatus.

Parameters	Parameter Value
Rotational speed of laser	Up to 3000 rpm
Drilling diameter	0.08 mm to 1.2 mm
Inclination angle	−0.7° to +0.7°

**Table 3 molecules-26-02953-t003:** Comparison of lasers with different pulse widths on the HAZ and drilling ability of hole cross-section topography with different drilling devices.

HAZ	Femtosecond Lasers	Picosecond Lasers	Nanosecond Lasers	Millisecond Laser
	<10 μm	<20 μm	<50 μm	Several Millimeters
Taper of hole cross section of CFRP	Drilling with spiral drilling apparatus	Scanning with galvanometer scanner	Several steps scanning with galvanometer scanner	Dual-beam laser drilling with galvanometer scanner
	Cylindrical, positive taper and inverted cones holes	Positive taper hole and cross-section taper > 1.25°	The bottom end of the hole is right angle, and the upper part of the hole still has a taper	The two ends of the hole are at right angles, and there is still a taper in the middle of the hole

## Data Availability

The data presented in this study are available in article.

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
