# Peer review of "Femtosecond Laser Drilling of Cylindrical Holes for Carbon Fiber-Reinforced Polymer (CFRP) Composites"

_molecules, 2021, doi:10.3390/molecules26102953_

Round 1

Reviewer 1 Report

The work presents an effort to machine quality holes in CFRP using a femtosecond laser. I believe there is a need to improve the current version. I have a few questions and comments.

  1. When you specify the femtosecond laser processing is better than picosecond laser processing then it is reasonable to show the clear improvements.
  2. The abstract fails to show with novelty and rigour. Please mention why your work is better with respect to pico laser processing.
  3. There have been several studies on femtosecond processing of CFRP which is missing in the introduction. The idea of introduction should show the motivation of the present research. Rewrite introduction to show the originality of the research.
  4. Please use appropriate units. Joule represents the laser energy not the power.
  5. Please include raw beam and focal spot diameters and what optics have been used.
  6. I think using the term 'drilling speed' is incorrect here as it is actually the rotational speed of the laser. It is good to mention the vertical speed as well (Z movement).
  7. How many holes were studied to see the parameter effect? There is no error bars so I believe only one hole which is unacceptable. This may be the reason behind such a behaviour for the studied laser powers.
  8. How would you address the anisotropic nature of CFRPs against laser drilling? 
  9. Please present clear SEM images of HAZ for the power and rotational speed comparison. The current images aren't up to the standard.
  10. The significance of the work lies in improving HAZ thickness. Please compare the results with other published work so that the readers know a clear benefit.

Reviewer 2 Report

The current manuscript presents the effect of femtosecond laser drilling of CFRP on the quality of the drilled hole. The title is attractive and highly recommended for specific applications, and the data presented are very interesting. However, some minor points should be addressed as follows:

  • The authors should refer to the direct applications of using the femtosecond laser drilling through the introduction and conclusion sections.
  • It is recommended to add a table comparing the current results to those obtained by the literature studies. This could verify the advantages of using the current technique.
  • Some figures need more explanation for their images such as Figures 6, 8, and 13.
  • The figures which display the measured data plot curves (such as figures2, 3, 7 and 9 ) don't present any standard deviation values, please add the standard deviation for each figure.
  • The bullet points in the conclusion section should be more concise and focused. 

Reviewer 3 Report

A commendable work by its content, but with many formatting issues. It seems that the authors were in a hurry when finishing it.

Please also provide a photo of the experimental layout.

Figure 1 is too straightforward, and the notations are too small, difficult to be read.

Line 80 and Table 1 (and in many other places in the paper): use W (capital letter) instead of w for Watts for power units.

Please consider a space between value and units (for example in Line 90 16 mm instead of 16mm).

What does “single ring shape” and “spiral ring” mean ? If these are toolpaths (relative trajectories between laser beam and part), please use the appropriate term “toolpath” of “(relative) processing trajectory”.

Also, please provide a graphical description (drawing) of these toolpaths, with all geometric and kinematic information.

Please use a space, like that “Fig. 1” instead of Fig.1 (without space). Do that for all figures !

Figures 8-10: the letters on figures are too small, making the figures unreadable.

Round 2

Reviewer 1 Report

The authors have revised the paper carefully addressing most of the comments raised. However, I still find a few mistakes. Please proof-read the manuscript.

  1. Page 1, line 37: What do you mean by non-contact stress processing?
  2. When you cite someone, it is a standard to write the reference number just after the name of the authors (e.g. XX et al [A]).
  3. Page 2, line 87: “90° lay-ups of carbon”. It should be carbon fibre.
  4. Page 2, line 90: “pulse power of 200 µJ”. change it to pulse energy. Proof-read for similar mistakes. Check Table 1.
